# Generalization Bounds for Canonicalization: A Comparative Study with Group Averaging

**Behrooz Tahmasebi**
MIT*
bzt@mit.edu

**Stefanie Jegelka**
TUM† and MIT*

## Abstract

Canonicalization, a popular method for generating invariant or equivariant function classes from arbitrary function sets, involves initial data projection onto a reduced input space subset, followed by applying any learning method to the projected dataset. Despite recent research on the expressive power and continuity of functions represented by canonicalization, its generalization capabilities remain less explored. This paper addresses this gap by theoretically examining the generalization benefits and sample complexity of canonicalization, comparing them with group averaging, another popular technique for creating invariant or equivariant function classes. Our findings reveal two distinct regimes where canonicalization may outperform or underperform compared to group averaging, with precise quantification of this phase transition in terms of sample size, group action characteristics, and a newly introduced concept of *alignment*. To the best of our knowledge, this study represents the first theoretical exploration of such behavior, offering insights into the relative effectiveness of canonicalization and group averaging under varying conditions.

## 1 Introduction

The goal of learning with invariances is to leverage known symmetries present in data to build models that are inherently invariant. Such symmetries frequently arise in various machine learning applications, particularly in the natural sciences (see, e.g., (Batzner et al., 2022; Grisafi et al., 2018; Unke et al., 2021)). Examples include Euclidean symmetries and equivariances (Smidt, 2021), among others. These forms of invariance are collectively addressed within the broader framework of geometric deep learning (Bronstein et al., 2017).

Several approaches are available for embedding invariances into machine learning models, including designing models with built-in invariance tailored to specific applications. Notable examples include Graph Neural Networks (GNNs) for graph data (Scarselli et al., 2008; Xu et al., 2019a), Convolutional Neural Networks (CNNs) for image data (Li et al., 2021; Krizhevsky et al., 2012), and PointNet for point clouds (Qi et al., 2017a;b). These methods rely on tailoring the network architecture to the particular type of invariance relevant to the application.

Another common approach to introducing invariances is to use a base function class and augment it with additional modules to ensure the final representation is invariant with respect to any group. Techniques in this category include group averaging (Murphy et al., 2019), frame averaging (Puny et al., 2022), and canonicalization (Kaba et al., 2023), the latter of which forms the central focus of this paper.

In canonicalization, the data is first mapped onto a lower-dimensional space to reduce redundancies arising from inherent invariances. A model, such as a neural network, is then trained on this transformed data (see Figure 2). This contrasts with group averaging, where the model is trained such that its output, averaged over all group transformations of the data, is plausible.

Given the empirical success of these methods for learning under invariance, there has been significant interest in understanding their theoretical foundations, particularly in terms of expressive power

---

*EECS and CSAIL.
†CIT, MCML, and MDSI.

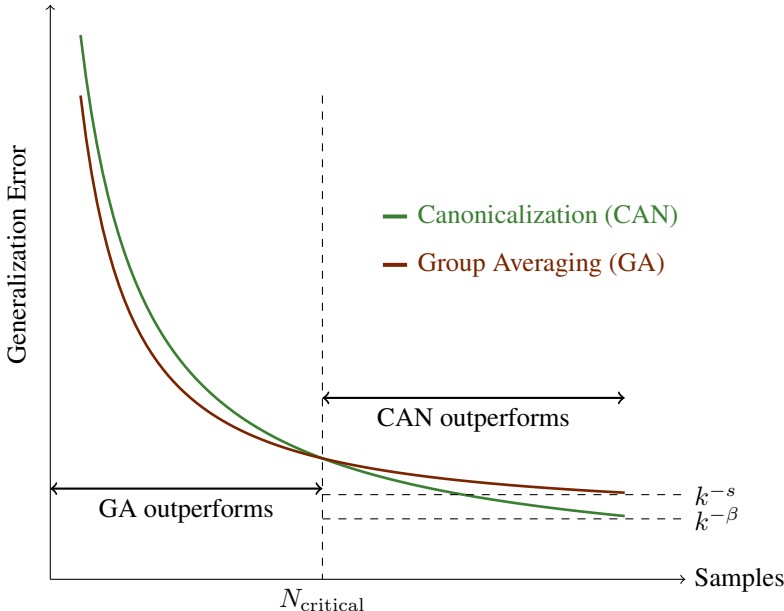

Figure 1: When $n \geq N_{\text{critical}}$, canonicalization demonstrates superior performance over group averaging in terms of generalization error (sample complexity). In this scenario, the optimal target function $f^\star$ aligns with the canonicalization scheme ($\beta > s$). Conversely, when $n \leq N_{\text{critical}}$, group averaging is preferred as it provides a smoother approach to constructing invariant functions.

and benefits like sample complexity and generalization bounds. For example, it has been shown that group averaging enables strong generalization, allowing models to learn effectively with much smaller sample sizes compared to cases without invariance. However, much less is known about the theoretical properties of canonicalization.

In this paper, we aim to address the gap in understanding the generalization properties of canonicalization by studying it compared to group averaging, which serves as a baseline. This joint analysis reveals the strengths of both approaches and provides a theoretical basis for their comparison. Specifically, we seek to answer the following question:

> Under what conditions do canonicalized models generalize well, and when do they outperform group averaging?

To this end, we introduce the concept of *alignment*. A target function is said to be aligned with a canonicalization scheme if it can be well-approximated by end-to-end canonicalized representations derived from "simple" base functions. This idea is inspired by *algorithmic alignment* (Xu et al., 2019b), which was developed to explain the generalization capabilities of Graph Neural Networks (GNNs) by demonstrating that they produce functions aligned with dynamic programming-like distributed algorithms on graphs. Furthermore, prior work has shown that learned canonicalization can be highly effective in various applications (Kaba et al., 2023). This paper provides a theoretical foundation for understanding how learned canonicalization may outperform group averaging when the canonicalization is more aligned with the downstream task.

In particular, we identify a phase transition when comparing the generalization performance of canonicalization and group averaging (Figure 1). Specifically, if the number of samples available to the learning algorithm is below a critical threshold, denoted as $N_{\text{critical}}$, group averaging exhibits superior performance. Conversely, when the number of samples exceeds $N_{\text{critical}}$, canonicalization has the potential to outperform group averaging, provided the canonicalized model is well-aligned with the target function (Definition 2). The improvement offered by canonicalization in such cases can grow arbitrarily large as the alignment with the target task increases. Furthermore, we derive a

complete characterization of $N_{\text{critical}}$ as a function of the properties of the group and the underlying function space.

To the best of our knowledge, this is the first theoretical investigation into this behavior, offering new insights into the comparative effectiveness of canonicalization and group averaging under different conditions.

In summary, this paper makes the following contributions:

- We analyze the generalization bounds and sample complexity advantages of learning with canonicalized models.

- We conduct a comparative theoretical study with group averaging to identify when canonicalization can outperform group averaging in terms of sample complexity.

- We introduce the notion of *alignment* for canonicalized models and demonstrate that the superiority of canonicalization or group averaging depends on whether the target task aligns with the canonicalization scheme. To our knowledge, this is the first detailed exploration of this behavior for both canonicalization and group averaging.

## 2 RELATED WORK

Euclidean symmetry and equivariance have garnered significant recent interest in machine learning applications (Smidt, 2021; Bronstein et al., 2017), although their study dates back to earlier works (Hinton, 1987; Batzner et al., 2023). These concepts have led to numerous approaches for incorporating symmetry into machine learning models, including techniques such as group averaging (Murphy et al., 2019), frame averaging (Puny et al., 2022), canonicalization (Kaba et al., 2023; Ma et al., 2024a; Panigrahi & Mondal, 2024), and random projections (Dym & Gortler, 2024). It is worth noting that canonicalization can face challenges related to discontinuities and stability issues (Dym et al., 2024).

Moreover, the study of invariances extends beyond learning; recent works have explored learning invariances in neural networks (Benton et al., 2020), measuring invariances (Goodfellow et al., 2009), and optimization approaches that account for invariances (Teo et al., 2007); see also (Bloem-Reddy et al., 2020; Chaimanowong & Zhu, 2024). Specific applications of canonicalization, particularly for sign and basis invariances, have also been recently proposed (Lim et al., 2023; 2024), with related work on Laplacian-based approaches (Ma et al., 2024b).

Consequently, the generalization capabilities of invariant classifiers have attracted significant attention in recent years (Sokolic et al., 2017). For learning with kernels (Scholkopf & Smola, 2018), several studies have explored group averaging to develop kernel methods for handling invariances, investigating its generalization error (Tahmasebi & Jegelka, 2023; Bietti et al., 2021; Elesedy, 2021; Mei et al., 2021).

For equivariance, numerous approaches have also been proposed, such as parameter sharing (Ravanbakhsh et al., 2017), with various works examining its generalization properties (Petrache & Trivedi, 2023; Behboodi et al., 2022; Elesedy & Zaidi, 2021). Incorporating equivariances in sampling for generative models has likewise been shown to be beneficial (Biloš & Günnemann, 2021; Köhler et al., 2020; Niu et al., 2020). Furthermore, recent work has explored the complexity of learning under invariances for gradient-based algorithms (Kiani et al., 2024).

## 3 PROBLEM STATEMENT

We consider a classical learning setup with a dataset of $n$ labeled examples, $\mathcal{S} = \left\{ (x_i, y_i) : i \in [n] \right\}$, where the inputs $x_i$, $i \in [n]$, are i.i.d. samples from a uniform distribution over the manifold input space $\mathcal{X} \subseteq \mathbb{R}^d$. The labels are generated by an *unknown* continuous target function $f^\star$, such that $y_i = f^\star(x_i) + \epsilon_i$, where the noise terms $\epsilon_i$, for $i \in [n]$, are independent, zero-mean random variables with variance bounded by $\sigma^2$. The objective is to learn $f^\star$ from the data.

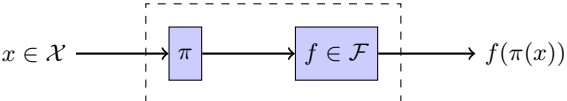

Figure 2: Canonicalization for creating invariant function classes.

Empirical Risk Minimization (ERM) provides an estimator for this problem through the following optimization program:

$$\min_{f \in \mathcal{F}} \frac{1}{n} \sum_{i \in [n]} \ell(f(x_i), y_i), \tag{1}$$

where $\ell(\cdot, \cdot)$ denotes the squared loss, and $\mathcal{F}$ represents the function class from which $f$ is selected.

### 3.1 LEARNING WITH INVARIANCES

In the context of learning with invariances, the optimal target function $f^\star$ is invariant under the continuous action of a group $G$ on the input space $\mathcal{X}$. This means that for all $x \in \mathcal{X}$ and $g \in G$, the functional equation $f^\star(gx) = f^\star(x)$ holds. Here, $gx$ denotes the transformation of the element $x \in \mathcal{X}$ under the action of the group element $g \in G$. For simplicity, we assume throughout this paper that $G$ is finite.

Note that the invariance of $f^\star$ does not necessarily imply that the ERM solution is an invariant estimator. To leverage these invariances in the learning process, various methods can be employed, such as group averaging, canonicalization, data augmentation, and frame averaging. In this paper, we focus on the first two methods and provide a brief review of these approaches in the following subsections.

### 3.2 GROUP AVERAGING

Group averaging is a method for obtaining invariant functions from a set of base functions $\mathcal{F}$. The idea is that given an arbitrary function $f \in \mathcal{F}$, the function $\mathcal{R}[f]$, defined as

$$\mathcal{R}[f](x) := \frac{1}{|G|} \sum_{g \in G} f(gx), \tag{2}$$

is invariant with respect to the group $G$. Here, $\mathcal{R}[\cdot]$ is sometimes referred to as the Reynolds operator corresponding to the group $G$. Given a function space $\mathcal{F}$, let us denote the space of functions represented via group averaging as $\mathcal{F}_{\mathrm{GA}}$ which is formally defined as follows:

$$\mathcal{F}_{\mathrm{GA}} := \Big\{ \mathcal{R}[f] : f \in \mathcal{F} \Big\}. \tag{3}$$

### 3.3 CANONICALIZATION

Canonicalization provides an alternative approach to constructing invariant function classes. Let $\mathcal{X}/G$ denote the quotient space of the action of $G$ on $\mathcal{X}$, formally defined as $\mathcal{X}/G := \{[x] : x \in \mathcal{X}\}$, where $[x] := \{gx : g \in G\}$ represents the orbits of the group. We assume $\mathcal{X}/G$ is embedded in the input space $\mathcal{X}$, meaning $\mathcal{X}/G \subseteq \mathcal{X}$. The embedding (or projection) map $\pi : \mathcal{X} \to \mathcal{X}/G$ maps elements from the input space to their representation in the canonicalized space $\mathcal{X}/G$.

Figure 2 illustrates how canonicalized function classes are constructed from a set of base functions $f \in \mathcal{F}$. In canonicalized models, any $x \in \mathcal{X}$ is first mapped to the canonicalized (or quotient) space through the projection function $\pi$. The projected data $\pi(x)$ is then passed to a base function $f \in \mathcal{F}$, yielding the final output $f(\pi(x))$. The set of all canonicalized functions is defined as:

$$\mathcal{F}_{\mathrm{CAN}} := \Big\{ f(\pi(x)) : f \in \mathcal{F} \Big\}. \tag{4}$$

### 3.4 SETUP AND ASSUMPTIONS

As discussed, in learning with invariances, we begin with a base function class $\mathcal{F}$ and then employ a method to construct invariant function classes, such as group averaging $\mathcal{F}_{\mathrm{GA}}$ or canonicalization $\mathcal{F}_{\mathrm{CAN}}$. We then formulate the problem as an empirical risk minimization (ERM) over the new invariant function class to achieve an invariant estimator.

It is well known that both function classes $\mathcal{F}_{\mathrm{GA}}$ and $\mathcal{F}_{\mathrm{CAN}}$ are universally expressive: under mild conditions on the set base functions $\mathcal{F}$, both function spaces can approximate *any* invariant and continuous function. However, universal expressiveness is *not* sufficient to ensure *learnability*. While expressiveness concerns to the ability of the new function class to approximate any continuous function, learnability concerns whether an algorithm (such as ERM) can learn the target function with appropriate sample and computational complexity while maintaining a small generalization error.

In this paper, we study the sample efficiency of canonicalization by comparing it with group averaging. We explicitly ask:

> Which invariant function class ($\mathcal{F}_{\mathrm{GA}}$ or $\mathcal{F}_{\mathrm{CAN}}$) requires a lower number of samples to generalize? Which one is more sample efficient?

We aim to provide a meaningful answer to this question in the next section.

*Assumptions.* In this paper, we focus of finite subgroups $G$ of the orthogonal group $O(d)$ acting on the input space $S^{d-1} := \{x \in \mathbb{R}^d : \|x\|_2 = 1\}$. Note that our goal is to learn an appropriate base function $f \in \mathcal{F}$ such that, after applying the necessary transformations (either group averaging or canonicalization), we obtain a highly accurate estimator of $f^\star$. Therefore, we assume that *learning* a base function $f \in \mathcal{F}$ is feasible. To formalize this, we avoid overly complex functions in the base space by defining the set of base functions as polynomials of $x \in S^{d-1}$ with degree at most $k \in \mathbb{N}$:

$$\mathcal{F}^k := \left\{ f \in L^2(S^{d-1}) : f \text{ is a polynomial of degree at most } k \right\}. \tag{5}$$

The parameter $k \in \mathbb{N}$ controls the complexity of the base space (i.e., the number of parameters to learn). For small $k$, the focus is on low-dimensional (i.e., smoother and simpler) functions, which are generally easier to learn. Let the invariant function classes for the base space $\mathcal{F}^k$ be denoted as $\mathcal{F}^k_{\mathrm{CAN}}$ and $\mathcal{F}^k_{\mathrm{GA}}$.

*Target function complexity.* We also need to control the complexity of the target function $f^\star \in L^2(S^{d-1})$. To achieve this, we adopt a common assumption from the literature, namely that $f^\star \in H^s(S^{d-1})$, where $H^s(S^{d-1})$ represents the Sobolev space with parameter $s \geq 0$. Formally, this space is defined as:

$$H^s(S^{d-1}) := \left\{ f \in L^2(S^{d-1}) : f \text{ has square-integrable derivatives (on the sphere) up to order } s \right\}.$$

It is well known that to obtain continuous functions, one must assume $s > (d-1)/2$, which we adopt in this work. Additionally, larger values of $s$ correspond to the integrability of higher-order derivatives, leading to smoother functions.

*On the restrictiveness of assumptions.* In this paper, we restrict our focus to the sphere and finite matrix group actions for simplicity. While the setup can be extended to a more general framework, we prioritize simplicity here. The choice of using low-degree polynomials for the base space is closely related to Sobolev kernels, which are commonly employed in machine learning and statistics. To obtain precise convergence rates for the generalization error (or excess population risk), we focus on low-degree functions, leveraging results from the theory of kernel convergence.

## 4 MAIN RESULTS

In this section, we study generalization bounds for the problem of learning under invariances via canonicalization, as discussed in the previous section. We begin by studying and comparing the expressive powers of canonicalization and group averaging.

### 4.1 WARM-UP: EXPLORING EXPRESSIVE POWER

Here, we compare the expressive power of $\mathcal{F}_{\text{CAN}}$ and $\mathcal{F}_{\text{GA}}$, with particular interest in cases where the class of base functions is *not* universally expressive (e.g., low-degree polynomials). Intuitively, we aim to use the next theorem to understand the expressive power for *learnable* functions, which constitute a significantly smaller subset of the continuous functions class. Furthermore, the following result also provides better insights before we present the main generalization bound of the paper.

**Theorem 1** (Expressive power of canonicalization). *Consider an arbitrary vector space of base functions $\mathcal{F}$, and assume it is closed under the group action, i.e., $f(gx) \in \mathcal{F}$ for each $f \in \mathcal{F}$ and $g \in G$. Then, it follows that $\mathcal{F}_{\text{CAN}} \supseteq \mathcal{F}_{\text{GA}}$.*

We present the proof of Theorem 1 in Appendix B.

The above result indicates that if the set of learnable functions satisfies mild conditions (i.e., being a vector space and closed under the group action), then canonicalization is *always* superior to group averaging in terms of *approximation error*. This implies that canonicalized models serve as better approximators of the target function. However, from basic learning theory, we know that while larger function classes correspond to smaller *bias*, they also incur larger *variance*. Specifically, if $f^\star \in \mathcal{F}_{\text{GA}}$ (or is close to lying within that space), then canonicalized models introduce more complexity into the learning task. In such cases, one might expect that

$$\text{Generalization Error of } \mathcal{F}_{\text{GA}} \ll \text{Generalization Error of } \mathcal{F}_{\text{CAN}}. \tag{6}$$

Conversely, if the canonicalized model *aligns* with the target function—meaning that $f^\star \in \mathcal{F}_{\text{CAN}}$ (or is close to lying within it), but $f^\star \notin \mathcal{F}_{\text{GA}}$—then one should expect that, for large sample sizes, the canonicalized model generalizes better due to its lower approximation error:

$$\text{Generalization Error of } \mathcal{F}_{\text{CAN}} \ll \text{Generalization Error of } \mathcal{F}_{\text{GA}}. \tag{7}$$

This highlights a dichotomy regarding whether canonicalization outperforms group averaging in terms of generalization and sample complexity. We will formalize this observation in the next subsection.

For the remainder of this subsection, we formally define the concept of alignment through a more concrete mathematical formulation in the context of approximation theory.

**Definition 2** (Alignment). A $G$-invariant target function $f^\star \in L^2(S^{d-1})$ is said to be $\beta$-aligned with the canonicalized models $\mathcal{F}_{\text{CAN}}^k$ if and only if

$$\min_{f \in \mathcal{F}_{\text{CAN}}^k} \|f - f^\star\|_{L^2(S^{d-1})} \leq Ck^{-\beta}, \tag{8}$$

for all $k \in \mathbb{N}$, where $C$ is an absolute constant that does not depend on $k$.

Observe that a larger value of $\beta$ corresponds to greater alignment with canonicalized functions. Indeed, as $\beta \to \infty$, the function becomes nearly aligned with canonicalized models even for finite $k$. For Sobolev target functions $f^\star \in H^s(S^{d-1})$, alignment is only non-trivial when $\beta > s$.

**Proposition 3.** *Any arbitrary $G$-invariant Sobolev function $f^\star \in H^s(S^{d-1})$ with parameter $s$ is $s$-aligned.*

The proof of Proposition 3 is provided in Appendix C and is based on an equivalent definition of Sobolev spaces discussed in Appendix A. According to this result, in the remainder of the paper, we will always consider cases where $\beta \geq s > (d-1)/2$.

To conclude this subsection, we provide an example to demonstrate that the inequality in Theorem 1 can be strict, meaning that there exist cases where $\mathcal{F}_{\text{GA}}$ is a *proper* subset of $\mathcal{F}_{\text{CAN}}$, i.e., $\mathcal{F}_{\text{CAN}} \supset \mathcal{F}_{\text{GA}}$ and $\mathcal{F}_{\text{CAN}} \neq \mathcal{F}_{\text{GA}}$.

**Example 4.** Consider the space of linear functions defined as:

$$\mathcal{F}^1 = \left\{ \sum_{i \in [d]} a_i x_i \mid \forall i \in [d] : a_i \in \mathbb{R} \right\}. \tag{9}$$

Let $P_d$ denote the group of all permutation matrices (i.e., the symmetric group) acting on $x \in S^{d-1}$. Note that $\mathcal{F}^1$ satisfies the condition stated in Theorem 1. We have

$$\mathcal{R}[f](x) = \frac{1}{d!} \sum_{\sigma \in P_d} f(\sigma x) = \sum_{i \in [d]} a x_i, \tag{10}$$

where $a = \frac{1}{d}\sum_{i\in[d]} a_i$. In other words, $\mathcal{F}_{\mathrm{GA}}^1$ is a one-dimensional function class consisting solely of linear functions of the form $a\sum_{i\in[d]} x_i$ for some $a \in \mathbb{R}$.

Now, let us consider canonicalized functions with respect to $P$ via the sort mapping: $\pi(x) := (x_{\min}, \ldots, x_{\max})^\top$. In this case, if we specifically consider functions restricted to $S^{d-1}/P_d$, then $\mathcal{F}_{\mathrm{CAN}}^1$ can represent *any* linear function $\sum_{i\in[d]} a_i x_i$ on the quotient space. In contrast, group averaging is only able to represent functions of the form $a\sum_{i\in[d]} x_i$, even when restricted to the quotient space. For a concrete example, consider the function $f(x) = \min_{i\in[d]} x_i$. This function belongs to $\mathcal{F}_{\mathrm{CAN}}^1$ but is not in $\mathcal{F}_{\mathrm{GA}}^1$ because it is not linear on $S^{d-1}$. This demonstrates that $\mathcal{F}_{\mathrm{CAN}}^1 \supset \mathcal{F}_{\mathrm{GA}}^1$ and $\mathcal{F}_{\mathrm{CAN}}^1 \neq \mathcal{F}_{\mathrm{GA}}^1$. The same holds for $\mathcal{F}^k$, for all $k \in \mathbb{N}$.

*Remark 5.* As vector spaces, note that $\dim(\mathcal{F}_{\mathrm{CAN}}^1) = d$ while $\dim(\mathcal{F}_{\mathrm{GA}}^1) = 1$. In other words, the gap between the two vector spaces can be *arbitrarily* large.

## 4.2 GENERALIZATION AND SAMPLE COMPLEXITY

In this section, we present the primary result of this paper, stated in the following theorem.

**Theorem 6** (Generalization bounds for canonicalization). *Consider the problem of learning under invariances with a dataset $\mathcal{S} = \{(x_i, y_i) : i \in [n]\} \subseteq (S^{d-1} \times \mathbb{R})^n$, consisting of $n$ i.i.d. (labeled) samples, where $y_i = f^\star(x_i) + \epsilon_i$ and the independent noise terms $\epsilon_i$ are zero-mean with variance bounded by $\sigma^2$ for all $i \in [n]$. Let $\widehat{f}_{\mathrm{CAN}}$ and $\widehat{f}_{\mathrm{GA}}$ denote the Empirical Risk Minimization (ERM) estimators derived from the invariant function classes $\mathcal{F}_{\mathrm{CAN}}^k$ and $\mathcal{F}_{\mathrm{GA}}^k$, respectively:*

$$\widehat{f}_{\mathrm{CAN}} = \arg\min_{f\in\mathcal{F}_{\mathrm{CAN}}^k} \frac{1}{n} \sum_{i\in[n]} \ell(f(x_i), y_i), \tag{11}$$

$$\widehat{f}_{\mathrm{GA}} = \arg\min_{f\in\mathcal{F}_{\mathrm{GA}}^k} \frac{1}{n} \sum_{i\in[n]} \ell(f(x_i), y_i), \tag{12}$$

*where $\ell(\cdot, \cdot)$ denotes the squared loss. Additionally, we assume the following conditions:*

- *The optimal target function $f^\star$ belongs to the Sobolev space $H^s(S^{d-1})$ with $s > \frac{d-1}{2}$.*

- *The function $f^\star$ is $\beta$-aligned with the canonicalization scheme for $\mathcal{F}_{\mathrm{CAN}}$, where $\beta \geq s$.*

*Under these assumptions, we obtain the following bounds with high probability[1]:*

$$\left\| \widehat{f}_{\mathrm{CAN}} - f^\star \right\|_{L^2(S^{d-1})}^2 \lesssim \widehat{L}_{\mathrm{CAN}} := \sum_{\ell=0}^{k} \dim(Y_{d,\ell}) \frac{\sigma^2}{n} + k^{-2\beta}, \tag{13}$$

$$\left\| \widehat{f}_{\mathrm{GA}} - f^\star \right\|_{L^2(S^{d-1})}^2 \lesssim \widehat{L}_{\mathrm{GA}} := \sum_{\ell=0}^{k} \dim(Y_{d,\ell}^G) \frac{\sigma^2}{n} + k^{-2s}, \tag{14}$$

*where $Y_{d,\ell}$ denotes the space of spherical harmonics of degree $\ell$ over $d$ variables (Appendix A), and $Y_{d,\ell}^G \subseteq Y_{d,\ell}$ denotes its projection onto the space of $G$-invariant polynomials.*

The proof of Theorem 6 can be found in Appendix D, and a comprehensive review of the theory of spherical harmonics is provided in Appendix A.

The generalization curves obtained for group averaging and canonicalization are illustrated in Figure 1. Let us interpret the upper bound in Theorem 6. First, note that according to Appendix A, we have the following bounds:

$$\sum_{\ell=0}^{k} \dim(Y_{d,\ell}) = \frac{2k^{d-1}}{(d-1)!}(1 + o_k(1)), \quad \sum_{\ell=0}^{k} \dim(Y_{d,\ell}^G) = \frac{2k^{d-1}}{|G|(d-1)!}(1 + o_k(1)), \tag{15}$$

where $o_k(1) \to 0$ as $k \to \infty$. This leads to the following corollary.

---

[1] In this paper, all high-probability arguments hold with at least 90% probability.

**Corollary 7.** *Let us define the critical sample complexity $N_{\text{critical}}$ as follows:*

$$N_{\text{critical}} := \frac{\sigma^2 k^{2s+d-1}}{(d-1)!(1-|G|^{-1})(1-k^{2(s-\beta)})}. \tag{16}$$

*Then, each of the following implications holds with high probability:*

$$n \lesssim N_{\text{critical}} \implies \widehat{L}_{\text{GA}} \le \widehat{L}_{\text{CAN}} \tag{17}$$

$$n \gtrsim N_{\text{critical}} \implies \widehat{L}_{\text{CAN}} \le \widehat{L}_{\text{GA}}. \tag{18}$$

*Note that in the absence of alignment (i.e., $\beta = s$), we have that $N_{\text{critical}} = \infty$ and thus the latter case cannot occur, meaning that the upper bound obtained for canonicalization cannot outperform group averaging.*

The proof of Corollary 7 can be found in Appendix E.

To intuitively understand this phenomenon, note that when $n \lesssim N_{\text{critical}}$, we can disregard the second term in the upper bound and focus solely on the first term, which is the dominating term. This first term quantifies the generalization error for the functions generated from $\mathcal{F}_{\text{CAN}}$ or $\mathcal{F}_{\text{GA}}$ without any *bias*. Since $\mathcal{F}_{\text{GA}}$ represents a relatively smaller vector space (as established in Theorem 1), we conclude that group averaging is superior in these situations (i.e., it has lower *variance*).

Another way to interpret this regime is to recognize that group averaging provides a smoother approach to achieving invariant functions. In contrast, canonicalization involves composing base functions with the canonicalization (or projection) map $\pi$, which can be discontinuous or non-smooth (see Section 2 for references).

On the other hand, when $n \gtrsim N_{\text{critical}}$, the first term in the upper bound becomes negligible, allowing us to compare only the second term. If $\beta > s$, we can conclude that canonicalization outperforms group averaging in terms of generalization error. Intuitively, this corresponds to cases where the canonicalization scheme is *aligned* with the optimal target function (i.e., $\beta > s$). This leads to the following conclusion:

> Canonicalization outperforms group averaging when the sample size is sufficiently large. In such cases, the marginal gain of canonicalization depends on the degree to which the optimal target function is *aligned* with the canonicalization scheme.

It is important to note that the alignment condition can be achieved either through an inductive bias or through learned canonicalization schemes. In both scenarios, our theory suggests that canonicalization is superior, a conclusion that is strongly supported by observations from previous tasks, such as those involving graph neural networks in machine learning.

*Remark* 8. The upper bound established in Theorem 6 is tight in a minimax sense, and we include a proof of its optimality within the proof of Theorem 6. To attain this bound, one can utilize kernel regression within the space of (spherical) polynomials of degree at most $k$. This approach allows for achieving the upper bound by employing quadratic optimization to determine the optimal coefficients of the polynomial within the ERM objective. We provide a brief review of this in the proof of Theorem 6 as well.

*Remark* 9. It is important to note that the dependence of $N_{\text{critical}}$ on the group size $|G|$ and the alignment parameter $\beta$ is limited and negligible. In other words, the threshold at which the phase transition occurs does not significantly depend on the group size or the level of alignment. However, improved alignment—strongly influenced by the parameter $\beta$, the group, and the canonicalization scheme—enhances the benefits of canonicalization in the large sample size regime.

## 5 EXPERIMENTS

We present proof-of-concept experiments in this section. The primary focus of this paper is to understand the generalization behavior of canonicalization and its comparison to group averaging. However, we aim to demonstrate that even in simple settings, alignment plays a crucial role in performance.

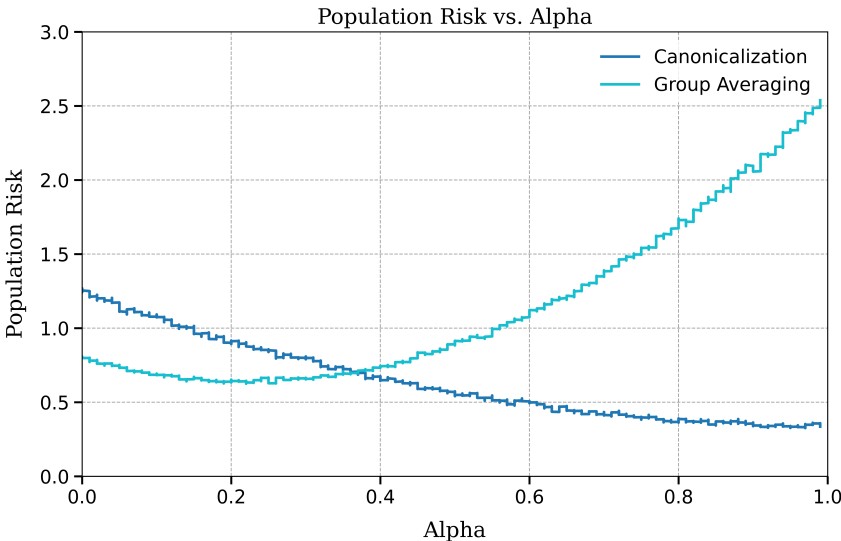

Figure 3: Generalization behavior of canonicalization compared to group averaging.

## 5.1 ALIGNMENT

Consider a linear model built on top of polynomial features of degree at most $k = 3$ over $d = 3$ dimensional uniform data from the cube $[-1, 1]^3$. The training data consists of $n = 100$ independent and identically distributed samples uniformly drawn from $[-1, 1]^3$, each labeled with the optimal target function $f^\star$ and corrupted by Gaussian noise with a standard deviation of $\sigma = 1$. We seek a permutation-invariant estimator $\widehat{f}$, either via group averaging or canonicalization, obtained through the least squares method.

The optimal target function, however, is assumed to be:

$$f^\star(x) = (1 - \alpha)\Big(x_1 + 2x_2 + 3x_3\Big) + \alpha\Big(\max(x_1, x_2, x_3) - 20\min(x_1, x_2, x_3)\Big), \quad (19)$$

where $\alpha \in [0, 1]$ is a hyperparameter. Note that as $\alpha \to 0$, $f^\star(x)$ becomes a linear but non-invariant function, making it difficult for both methods to learn. In this scenario, group averaging performs better, as it at least interpolates a smooth function rather than dealing with the non-smooth combination of $\max / \min$. This case lacks alignment, and we expect group averaging to outperform canonicalization.

On the other hand, as $\alpha \to 1$, we have $f^\star(x) = \max(x_1, x_2, x_3) - 20\min(x_1, x_2, x_3)$. This function is better aligned with canonicalization rather than group averaging. The reason is clear: it is already expressed in terms of $\max / \min$ functions, which are easier for canonicalization to manage. In other words, the function is more aligned with canonicalization.

These intuitions are consistent with the results we present in Figure 3. In this figure, we run experiments based on the above setting with $n = 100$ training and test samples, and noise level $\sigma = 1$. The results clearly show how alignment influences the distinct performance between group averaging and canonicalization.

## 5.2 POINT CLOUDS

In this section, we conduct an additional experiment to examine the theoretical results on the phase transition in the sample complexity of canonicalization and group averaging.

We use the following setup: assume a training set of $n$ point clouds, each consisting of $m$ points in $d$-dimensional Euclidean space, modeled as elements of $[-1, 1]^{m \times d}$. The point clouds are i.i.d. uniform samples from $[-1, 1]^{m \times d}$. The optimal target function is assumed to be $f^\star(x) = \max_{\ell \in [k]} \|x_\ell\|_1$, where $x_\ell$ denotes the $\ell$-th row of the point cloud $x \in [-1, 1]^{m \times d}$ (i.e., the points within the point cloud).

This target function is invariant to permutations of the points in the point cloud. Let $\sigma = 0.1$ denote the standard deviation of the Gaussian noise added to the training labels.

We train a two-layer ReLU network with a width of 20 on this dataset using mean-squared loss. Training is performed with SGD, using a learning rate of $0.01$ for 100 epochs. Two methods are compared: group averaging (over all $m!$ permutations of the point clouds) and canonicalization (via lexicographic sorting of the point cloud rows). The test loss is calculated over 100 uniformly random point clouds sampled from $[-1, 1]^{m \times d}$ as the test set.

For this experiment, we set $m = d = 5$ and report the average test loss along with the standard deviation over ten runs for different numbers of samples $n$, as shown below:

|  | $n = 10$ | $n = 100$ | $n = 1000$ |
|---|---|---|---|
| Group Averaging | $\mathbf{0.425^{\pm 0.132}}$ | $0.367^{\pm 0.076}$ | $0.365^{\pm 0.094}$ |
| Canonicalization | $0.471^{\pm 0.103}$ | $\mathbf{0.290^{\pm 0.061}}$ | $\mathbf{0.280^{\pm 0.059}}$ |

Table 1: Final test loss averaged over ten different random seeds.

The results reveal that when the number of samples $n$ is relatively small, group averaging outperforms canonicalization, though the test loss remains relatively high for both methods. However, as the number of samples increases, canonicalization achieves better test loss. This behavior aligns with the theoretical findings of this paper and provides empirical support for them.

# 6 CONCLUSION

In this paper, we investigate generalization bounds for canonicalization as a method for constructing invariant function classes. As a baseline, we compare it with group averaging, a widely used approach for generating invariant functions. Our findings reveal two distinct regimes where canonicalization either outperforms or underperforms group averaging, in terms of the convergence of generalization error (i.e., sample complexity). Specifically, in the low-sample regime, group averaging proves superior due to its smoother effect on the base function class. However, in the high-sample regime, canonicalization performs better, as it leverages the canonicalization (or projection) map to achieve more accurate model approximations. This highlights the importance of aligning the canonicalization process with the target task, which we demonstrate to be crucial for building models with strong generalization capabilities. These results align with previous work in machine learning, particularly in areas like graph neural networks, where algorithmic alignment has been shown to enhance generalization.

ACKNOWLEDGMENTS

This work was supported by NSF AI Institute TILOS, ONR grant N00014-20-1-2023 (MURI ML-SCOPE), and an Alexander von Humboldt fellowship.

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

## A    BACKGROUND ON SPHERICAL HARMONICS

In this section, we review the theory of spherical harmonics, which provides a solid mathematical framework for understanding harmonic analysis on the sphere. This framework will be crucial for the proofs presented later in the paper.

Let $S^{d-1} := \{x \in \mathbb{R}^d : \|x\|_2 = 1\}$ denote the unit sphere embedded in $\mathbb{R}^d$. For any continuous function $f \in C(S^{d-1})$, we define its radial extension on $\mathbb{R}^d \setminus \{0\}$ as follows:

$$\widetilde{f}(x) := f\left(\frac{x}{\|x\|_2}\right).$$

This extension allows us to analyze the behavior of $f$ not just on the sphere but also in the whole space $\mathbb{R}^d$.

The spherical Laplacian, denoted by $\Delta_{S^{d-1}}$, is defined as:

$$\Delta_{S^{d-1}} f := \Delta \widetilde{f},$$

where $\Delta := \sum_{i=1}^{d} \partial_i^2$ represents the Euclidean Laplacian operator acting in $\mathbb{R}^d$. The operator $\Delta_{S^{d-1}}$ plays a pivotal role in spherical harmonic analysis.

A significant property of the spherical Laplacian $\Delta_{S^{d-1}}$ is that it is a self-adjoint operator on $L^2(S^{d-1})$. This self-adjointness ensures that the eigenvalues are real, and the eigenfunctions corresponding to these eigenvalues are orthogonal in the $L^2$ sense.

To explore the spectral properties of $\Delta_{S^{d-1}}$, we consider the vector space $V_\lambda$ consisting of solutions to the following partial differential equation (PDE):

$$\Delta_{S^{d-1}} \phi + \lambda \phi = 0,$$

defined on the unit sphere. A remarkable result is that the dimension of this vector space is finite:

$$\dim(V_\lambda) < \infty \quad \text{for all } \lambda \in \mathbb{R}.$$

Moreover, the dimension of $V_\lambda$ is non-zero if and only if $\lambda$ takes the specific form:

$$\lambda = \ell(\ell + d - 2)$$

for some non-negative integer $\ell$. This indicates that the eigenvalues of $\Delta_{S^{d-1}}$ are discrete and non-negative, forming a countable set.

The corresponding eigenfunctions associated with these eigenvalues constitute an orthonormal basis for the space $L^2(S^{d-1})$. These eigenfunctions, known as spherical harmonics, are crucial in various applications, including solving partial differential equations, performing expansions in spherical coordinates, and analyzing functions on the sphere.

The space $V_\lambda$ denotes the set of spherical harmonics corresponding to the eigenvalue $\lambda = \ell(\ell + d - 2)$. We let $\phi_{\ell,\ell'}$ with $\ell' \in [\dim(V_\lambda)]$ represent an orthogonal basis of this space. In order to compute the dimension of $V_\lambda$, we refer to the established theory of spherical harmonics.

We know that the space $V_\lambda$ can be expressed as:

$$V_\lambda = \Big\{ h(x) : h \text{ is a homogeneous harmonic polynomial of degree } \ell \text{ with } \lambda = \ell(\ell + d - 2) \Big\},$$

where a harmonic polynomial $h : \mathbb{R}^d \to \mathbb{R}$ is defined as a polynomial that satisfies the condition $\Delta h = 0$, with $\Delta$ being the Laplacian operator.

Let $\mathcal{P}_\ell(\mathbb{R}^d)$ denote the ring of homogeneous polynomials of degree $\ell$ in $d$ variables. It can be shown that for any non-negative integer $\ell$, the relationship between the space of homogeneous polynomials and the space of spherical harmonics can be expressed as:

$$\mathcal{P}_\ell(\mathbb{R}^d) = V_{\ell(\ell+d-2)} \bigoplus r^2 \mathcal{P}_{\ell-2}(\mathbb{R}^d), \tag{20}$$

where $r^2 := x_1^2 + x_2^2 + \ldots + x_d^2$ is the radial polynomial. This decomposition reveals that the space of homogeneous polynomials can be broken down into the direct sum of spherical harmonics and a subspace generated by the radial component multiplied by lower-degree homogeneous polynomials.

Utilizing standard combinatorial arguments, we can derive the following corollary regarding the dimension of the space of spherical harmonics.

**Corollary 10.** *Let $Y_{d,\ell}$ denote the vector space of spherical harmonics of degree $\ell$. Specifically, we define $Y_{d,\ell}$ as $V_{\ell(\ell+d-2)}$. Then, the dimension of this space can be computed as:*

$$\dim(Y_{d,\ell}) = \binom{\ell+d-1}{d-1} - \binom{\ell+d-3}{d-1}. \tag{21}$$

This result provides a clear count of the dimensions of spherical harmonics in $d$-dimensional space, illustrating the rich structure of these functions. Moreover, it allows us to derive the dimension of spherical harmonics of degree at most $k \in \mathbb{N}$ in the following lemma.

**Lemma 11.** *We have the following asymptotic bounds for the dimension of spherical harmonics:*

$$\sum_{\ell=0}^{k} \dim(Y_{d,\ell}) = \frac{2k^{d-1}}{(d-1)!}(1 + o_k(1)), \tag{22}$$

*where $o_k(1) \to 0$ as $k \to \infty$.*

*Proof.* To establish this result, we begin by computing the sum of dimensions for spherical harmonics of degrees from 0 to $k$:

$$\sum_{\ell=0}^{k} \dim(Y_{d,\ell}) = \sum_{\ell=0}^{k} \left( \binom{\ell+d-1}{d-1} - \binom{\ell+d-3}{d-1} \right) \tag{23}$$

$$= \binom{k+d-1}{d-1} + \binom{k+d-2}{d-1}. \tag{24}$$

Thus, for large values of $k$,

$$\sum_{\ell=0}^{k} \dim(Y_{d,\ell}) = \binom{k+d-1}{d-1} + \binom{k+d-2}{d-1} \tag{25}$$

$$= \frac{k^{d-1}}{(d-1)!}(1 + o_k(1)) + \frac{k^{d-1}}{(d-1)!}(1 + o_k(1)) \tag{26}$$

$$= \frac{2k^{d-1}}{(d-1)!}(1 + o_k(1)). \tag{27}$$

This completes the proof. $\square$

Let us now compute the dimension of the vector space of spherical harmonics $Y_{d,\ell}^{G}$ that remain invariant under the action of a subgroup $G$ of the orthogonal matrices acting on $S^{d-1}$. According to the spectral theorems for invariant functions (Tahmasebi & Jegelka, 2023), we can derive the following result:

$$\sum_{\ell=0}^{k} \dim(Y_{d,\ell}^{G}) = \frac{(1 + o_k(1))}{|G|} \sum_{\ell=0}^{k} \dim(Y_{d,\ell}) = \frac{2k^{d-1}}{|G|(d-1)!}(1 + o_k(1)). \tag{28}$$

Spherical harmonics also play a crucial role in our analysis of polynomial regressions throughout this paper. Specifically, the direct sum $\bigoplus_{\ell=0}^{k} Y_{d,\ell}$ corresponds precisely to the space of polynomials of degree at most $k$ when we restrict our attention to the unit sphere.

We conclude this section by introducing the space of Sobolev functions $H^s(S^{d-1})$ in the context of spherical harmonics. Let $f \in L^2(S^{d-1})$ be a square-integrable function. From the properties associated with spherical harmonics, we know that $f$ can be expressed as an infinite series:

$$f(x) = \sum_{\ell=0}^{\infty} \sum_{\ell'=1}^{\dim(Y_{d,\ell})} f_{\ell,\ell'} \phi_{\ell,\ell'}(x), \tag{29}$$

where $\phi_{\ell,\ell'}(x)$, with $\ell = 0, 1, \ldots$ and $\ell' \in [\dim(V_\lambda)]$, form an orthonormal basis for both spherical harmonics and $L^2(S^{d-1})$. Indeed, these functions are homogeneous polynomials of degree $\ell$ as $\ell'$ varies.

Since $f \in L^2(S^{d-1})$, we can infer that

$$\sum_{\ell=0}^{\infty} \sum_{\ell'=1}^{\dim(Y_{d,\ell})} f_{\ell,\ell'}^2 = \|f\|_{L^2(S^{d-1})}^2 < \infty. \tag{30}$$

The concept of Sobolev spaces involves placing restrictions on this space by imposing conditions on the rate at which the tail of the above series vanishes. For any non-negative $s$, we define

$$H^s(S^{d-1}) := \Big\{ f \in L^2(S^{d-1}) : \|f\|_{H^s(S^{d-1})}^2 := \|f\|_{L^2(S^{d-1})}^2 \tag{31}$$

$$+ \sum_{\ell=0}^{\infty} \sum_{\ell'=1}^{\dim(Y_{d,\ell})} \ell^s (\ell + d - 2)^s f_{\ell,\ell'}^2 < \infty \Big\}. \tag{32}$$

One can prove that this new formulation is equivalent to the definition of Sobolev spaces presented in the main body of the paper.

## B    PROOF OF THEOREM 1

*Proof.* Let $\mathcal{F}$ be an arbitrary vector space closed under the group action. We want to prove that $\mathcal{F}_{\mathrm{CAN}} \supseteq \mathcal{F}_{\mathrm{GA}}$. To this end, fix an arbitrary $f \in \mathcal{F}$ and note that $\mathcal{R}[f] \in \mathcal{F}_{\mathrm{GA}}$. Our goal is to show that $\mathcal{R}[f] \in \mathcal{F}_{\mathrm{CAN}}$ and this completes the proof.

Note that for each $g \in G$, according to the closeness of $\mathcal{F}$ under group action, we have $f(gx) \in \mathcal{F}$. Moreover, since $\mathcal{F}$ is a vector space, and thus being closed under taking summations, we have that

$$\frac{1}{|G|} \sum_{g \in G} f(gx) \in \mathcal{F} \implies \mathcal{R}[f](x) \in \mathcal{F}. \tag{33}$$

Now let $\tilde{f} := \mathcal{R}[f](x)$. Then, according to the definition, we have that

$$\mathcal{R}[f](x) \in \mathcal{F} \implies \mathcal{R}[f](\pi(x)) \in \mathcal{F}_{\mathrm{CAN}}. \tag{34}$$

However, $\mathcal{R}[f](x) \in \mathcal{F}_{\mathrm{GA}}$ is $G$-invariant, thus we have

$$\mathcal{R}[f](x) = \mathcal{R}[f](\pi(x)), \tag{35}$$

for all $x$, which means that

$$\mathcal{R}[f](\pi(x)) \in \mathcal{F}_{\mathrm{CAN}} \implies \mathcal{R}[f](x) \in \mathcal{F}_{\mathrm{CAN}}, \tag{36}$$

and this completes the proof.

$\square$

## C    PROOF OF PROPOSITION 3

*Proof.* Let $f^\star \in H^s(S^{d-1})$ be a $G$-invariant function. To prove that $f^\star$ is $\beta$-aligned with the canonicalized models $\mathcal{F}_{\mathrm{CAN}}^k$ with $\beta = s$, we need to show that

$$\min_{f \in \mathcal{F}_{\mathrm{CAN}}^k} \|f - f^\star\|_{L^2(S^{d-1})} \le C k^{-s}, \tag{37}$$

for all $k \in \mathbb{N}$, where $C$ is an absolute constant that does not depend on $k$.

We begin by using Theorem 1, which implies that $\mathcal{F}_{\mathrm{CAN}}^k \supseteq \mathcal{F}_{\mathrm{GA}}^k$ for each $k \in \mathbb{N}$. The condition in Theorem 1 holds here because $\mathcal{F}^k$, the space of harmonic polynomials of degree at most $k$, is a vector space that is closed under group action.

Thus, we have

$$\min_{f \in \mathcal{F}_{\text{CAN}}^k} \|f - f^\star\|_{L^2(S^{d-1})} \leq \min_{f \in \mathcal{F}_{\text{GA}}^k} \|f - f^\star\|_{L^2(S^{d-1})}, \tag{38}$$

for each $k \in \mathbb{N}$. Therefore, the proof is complete if we show that

$$\min_{f \in \mathcal{F}_{\text{GA}}^k} \|f - f^\star\|_{L^2(S^{d-1})} \leq Ck^{-s}, \tag{39}$$

for each $k \in \mathbb{N}$.

Let $\Pi_k$ denote the orthogonal projection operator that projects square-integrable functions $f \in L^2(S^{d-1})$ to homogeneous harmonic polynomials (i.e., spherical harmonics) of degree at most $k \in \mathbb{N}$. We claim that $\Pi_k[f^\star] \in \mathcal{F}_{\text{GA}}^k$ and that

$$\left\| \Pi_k[f^\star] - f^\star \right\|_{L^2(S^{d-1})}^2 \leq C^2 k^{-2s}$$

for a constant $C$, which completes the proof.

First, since the Reynolds operator $\mathcal{R}$ commutes with $\Pi_k$, we have

$$\mathcal{R}[\Pi_k[f^\star]] = \Pi_k[\mathcal{R}[f^\star]] = \Pi_k[f^\star], \tag{40}$$

where we use $\mathcal{R}[f^\star] = f^\star$ since $f^\star$ is $G$-invariant. This shows that $\Pi_k[f^\star] \in \mathcal{F}_{\text{GA}}^k$ by definition.

Next, we show that $\left\| \Pi_k[f^\star] - f^\star \right\|_{L^2(S^{d-1})}^2 \leq C^2 k^{-2s}$. Let

$$f^\star(x) = \sum_{\ell=0}^{\infty} \sum_{\ell'=1}^{\dim(Y_{d,\ell})} f_{\ell,\ell'}^\star \phi_{\ell,\ell'}(x),$$

where $\phi_{\ell,\ell'}(x)$ (for $\ell = 0, 1, \dots$ and $\ell' \in [\dim(Y_{d,\ell})]$) form an orthonormal basis for both spherical harmonics and $L^2(S^{d-1})$. Then we have

$$\left\| \Pi_k[f^\star] - f^\star \right\|_{L^2(S^{d-1})}^2 = \sum_{\ell=k+1}^{\infty} \sum_{\ell'=1}^{\dim(Y_{d,\ell})} f_{\ell,\ell'}^2.$$

Define $\mu_\ell := \ell^s(\ell + d - 2)^s$ for each $\ell \in \mathbb{N}$. We obtain

$$\left\| \Pi_k[f^\star] - f^\star \right\|_{L^2(S^{d-1})}^2 = \sum_{\ell=k+1}^{\infty} \sum_{\ell'=1}^{\dim(Y_{d,\ell})} (f_{\ell,\ell'}^\star)^2 \tag{41}$$

$$= \sum_{\ell=k+1}^{\infty} \sum_{\ell'=1}^{\dim(Y_{d,\ell})} \mu_\ell^{-1} \mu_\ell (f_{\ell,\ell'}^\star)^2 \tag{42}$$

$$\leq \mu_k^{-1} \sum_{\ell=k+1}^{\infty} \sum_{\ell'=1}^{\dim(Y_{d,\ell})} \mu_\ell (f_{\ell,\ell'}^\star)^2 \tag{43}$$

$$= k^{-s}(k + d - 2)^{-s} \sum_{\ell=k+1}^{\infty} \sum_{\ell'=1}^{\dim(Y_{d,\ell})} \mu_\ell (f_{\ell,\ell'}^\star)^2. \tag{44}$$

Since $\mu_\ell$ is increasing in $\ell$, it follows that

$$\left\| \Pi_k[f^\star] - f^\star \right\|_{L^2(S^{d-1})}^2 \leq k^{-s}(k + d - 2)^{-s} \sum_{\ell=k+1}^{\infty} \sum_{\ell'=1}^{\dim(Y_{d,\ell})} \mu_\ell (f_{\ell,\ell'}^\star)^2 \tag{45}$$

$$\leq k^{-2s} \sum_{\ell=k+1}^{\infty} \sum_{\ell'=1}^{\dim(Y_{d,\ell})} \mu_\ell (f_{\ell,\ell'}^\star)^2 \tag{46}$$

$$\leq k^{-2s} \sum_{\ell=0}^{\infty} \sum_{\ell'=1}^{\dim(Y_{d,\ell})} \mu_\ell (f_{\ell,\ell'}^\star)^2 \tag{47}$$

$$\leq k^{-2s} \|f^\star\|_{H^s(S^{d-1})}^2. \tag{48}$$

This completes the proof.

$$\square$$

# D    PROOF OF THEOREM 6

*Proof.* In this section, we present the proof of Theorem 6. First, we use a closed-form formula for the ERM estimator, then decompose the error into its bias and variance components. Finally, we apply the analysis of random design linear regression to derive the result.

Consider a dataset $\mathcal{S} = \left\{ (x_i, y_i) : i \in [n] \right\} \subseteq \left( S^{d-1} \times \mathbb{R} \right)^n$, consisting of $n$ i.i.d. labeled samples, where $y_i = f^\star(x_i) + \epsilon_i$ and the independent noise terms $\epsilon_i$ are zero-mean with variance bounded by $\sigma^2$ for all $i \in [n]$. Let

$$\widehat{f}_{\text{CAN}} = \arg\min_{f \in \mathcal{F}_{\text{CAN}}^k} \frac{1}{n} \sum_{i \in [n]} \ell(f(x_i), y_i), \tag{49}$$

$$\widehat{f}_{\text{GA}} = \arg\min_{f \in \mathcal{F}_{\text{GA}}^k} \frac{1}{n} \sum_{i \in [n]} \ell(f(x_i), y_i), \tag{50}$$

where $\ell(\cdot, \cdot)$ denotes the squared loss. We aim to prove that for sufficiently large $k \in \mathbb{N}$, with high probability:

$$\left\| \widehat{f}_{\text{CAN}} - f^\star \right\|_{L^2(S^{d-1})}^2 \lesssim \sum_{\ell=0}^k \dim(Y_{d,\ell}) \frac{\sigma^2}{n} + k^{-2\beta}, \tag{51}$$

$$\left\| \widehat{f}_{\text{GA}} - f^\star \right\|_{L^2(S^{d-1})}^2 \lesssim \sum_{\ell=0}^k \dim(Y_{d,\ell}^G) \frac{\sigma^2}{n} + k^{-2s}. \tag{52}$$

First, we focus on group averaging. We need to introduce some notation, assuming the background provided in Appendix A. For any estimator, we have

$$f(x) = \sum_{\ell=0}^k \sum_{\ell'=1}^{\dim(Y_{d,\ell}^G)} f_{\ell,\ell'} \phi_{\ell,\ell'}^G(x),$$

where $\phi_{\ell,\ell'}^G(x)$ denotes a basis for $G$-invariant spherical harmonics. We can rewrite the empirical risk minimization (ERM) loss function as follows:

$$\frac{1}{n} \sum_{i \in [n]} \ell(f(x_i), y_i) = \frac{1}{n} \sum_{i=1}^n \left( \sum_{\ell=0}^k \sum_{\ell'=1}^{\dim(Y_{d,\ell}^G)} f_{\ell,\ell'} \phi_{\ell,\ell'}^G(x_i) - y_i \right)^2. \tag{53}$$

For simplicity in notation, we flatten our indices to $t := (\ell, \ell') \in [p]$, where $p := \sum_{\ell=0}^k \dim(Y_{d,\ell}^G)$. This avoids unnecessary dependence on previous notation for multiplicities.

Next, we introduce the feature matrix $\Phi = \left( \phi_t^G(x_i) \right)_{n \times p} \in \mathbb{R}^{n \times p}$. With a slight abuse of notation, we use $f$ to denote both a candidate estimator and its spherical coefficients. The ERM objective is:

$$\frac{1}{n} \sum_{i \in [n]} \ell(f(x_i), y_i) = \frac{1}{n} \| \Phi f - y \|_2^2, \quad f := \left( f_t \right)_{t \in [p]} \in \mathbb{R}^p. \tag{54}$$

According to the closed-form solution of the above least-squares objective, we obtain the following ERM estimator:

$$\widehat{f} = \Sigma^{-1} \Phi^\top y, \quad \Sigma := \Phi^\top \Phi, \tag{55}$$

assuming $\Sigma$ is invertible, which is equivalent to $n \geq p$ in our case.

With a slight abuse of notation, we also use $f^\star$ to denote the optimal target function and its optimal spherical coefficients. Replacing this into the population risk, we obtain:

$$\left\| \widehat{f}_{\text{GA}} - f^\star \right\|_{L^2(S^{d-1})}^2 = \| \widehat{f} - f^\star \|_2^2 \leq 2\| \widehat{f} - \Pi_k[f^\star] \|_2^2 + 2\| f_{>k}^\star \|_2^2, \tag{56}$$

where $\Pi_k$ denotes the orthogonal projection operator onto the space of spherical harmonics of degree at most $k$. We will treat the two terms on the right-hand side of the above inequality separately.

Let us first focus on the first term. Let $f_k^\star := \Pi_k[f^\star]$ and note that $f^\star = f_k^\star + f_{<k}^\star$. Moreover, $y = \Phi f_k^\star + \delta + \epsilon$, where $\delta_i := f_{<k}^\star(x_i)$ for any $i \in [n]$.

To obtain upper bounds on the first term, we utilize the derivation of $\widehat{f}$ and write:

$$\|\widehat{f} - \Pi_k[f^\star]\|_2^2 = \|\Sigma^{-1}\Phi^\top y - f_k^\star\|_2^2 \tag{57}$$

$$= \|\Sigma^{-1}\Phi^\top \Phi f_k^\star + \Sigma^{-1}\Phi^\top \delta + \Sigma^{-1}\Phi^\top \epsilon - f_k^\star\|_2^2 \tag{58}$$

$$\stackrel{(a)}{=} \|\Sigma^{-1}\Phi^\top \delta + \Sigma^{-1}\Phi^\top \epsilon\|_2^2 \tag{59}$$

$$\leq 2\|\Sigma^{-1}\Phi^\top \delta\|_2^2 + 2\|\Sigma^{-1}\Phi^\top \epsilon\|_2^2, \tag{60}$$

where (a) follows from the definition of $\Sigma$.

We note that

$$\mathbb{E}_\epsilon[\|\Sigma^{-1}\Phi^\top \epsilon\|_2^2] = \sigma^2 \mathrm{tr}(\Phi\Sigma^{-1}\Sigma^{-1}\Phi^\top) \tag{61}$$

$$= \sigma^2 \mathrm{tr}(\Phi^\top \Phi \Sigma^{-1}\Sigma^{-1}) \tag{62}$$

$$= \sigma^2 \mathrm{tr}(\Sigma^{-1}), \tag{63}$$

where in the last step, we used the definition of $\Sigma$. By a straightforward application of the matrix Bernstein inequality (Bach, 2024, Proposition 3.12), we obtain that $n\Sigma^{-1} \preceq I/4$ with high probability if $n \gtrsim p\log(p)$. Using Markov's inequality, this leads us to

$$\|\Sigma^{-1}\Phi^\top \epsilon\|_2^2 \lesssim \frac{\sigma^2 p}{n}, \quad \text{with probability at least } 99\%. \tag{64}$$

Next, we derive upper bounds on $\|\Sigma^{-1}\Phi^\top \delta\|_2^2$. Note that

$$\mathbb{E}_x[\|\Sigma^{-1}\Phi^\top \delta\|_2^2] = \mathbb{E}[\mathrm{tr}(\delta^\top \Phi\Sigma^{-1}\Sigma^{-1}\Phi^\top \delta)] \tag{65}$$

$$= \mathbb{E}[\mathrm{tr}(\delta\delta^\top \Phi\Sigma^{-1}\Sigma^{-1}\Phi^\top)] \tag{66}$$

$$\leq \mathbb{E}[\|\delta\delta^\top\|_F \, \|\Phi\Sigma^{-1}\Sigma^{-1}\Phi^\top\|_F] \tag{67}$$

$$\leq \mathbb{E}[\|\delta\delta^\top\|_F \, \mathrm{tr}(\Phi\Sigma^{-1}\Sigma^{-1}\Phi^\top)], \tag{68}$$

where $\|\cdot\|_F$ denotes the Frobenius norm. Note that $\|\delta\delta^\top\|_F = \|\delta\|_2^2$. Moreover, $\mathrm{tr}(\Phi\Sigma^{-1}\Sigma^{-1}\Phi^\top) = \mathrm{tr}(\Sigma^{-1})$. Thus, we have

$$\mathbb{E}_x[\|\Sigma^{-1}\Phi^\top \delta\|_2^2] \leq \mathbb{E}[\|\delta\|_2^2 \, \mathrm{tr}(\Sigma^{-1})]. \tag{69}$$

Similar to the previous case, we know that with high probability, $n\Sigma^{-1} \preceq I/4$ if $n \gtrsim p\log(p)$. Also,

$$\mathbb{E}[\frac{1}{n}\|\delta\|_2^2] = \mathbb{E}[\frac{1}{n}\sum_{i=1}^n f_{<k}^\star(x_i)^2] = \|f_{<k}^\star\|_2^2. \tag{70}$$

This allows us to conclude that:

$$\|\Sigma^{-1}\Phi^\top \delta\|_2^2 \lesssim \|f_{<k}^\star\|_2^2, \quad \text{with probability at least } 99\%. \tag{71}$$

We are now ready to combine all the above high-probability arguments to get:

$$\left\|\widehat{f}_{\mathrm{GA}} - f^\star\right\|_{L^2(S^{d-1})}^2 \lesssim \frac{\sigma^2 p}{n} + \|f_{<k}^\star\|_2^2, \quad \text{with probability at least } 99\%. \tag{72}$$

Now let us compute $\|f_{<k}^\star\|_2^2$. Similar to the proof of Proposition 3, we obtain

$$\|f_{<k}^\star\|_2^2 \leq k^{-2s}\|f^\star\|_{H^s(S^{d-1})}^2. \tag{73}$$

Therefore, we have

$$\left\|\widehat{f}_{\mathrm{GA}} - f^\star\right\|_{L^2(S^{d-1})}^2 \lesssim \frac{\sigma^2 p}{n} + k^{-2s}, \quad \text{with probability at least } 90\%, \tag{74}$$

which completes the proof.

*Remark* 12. The proof for $\mathcal{F}_{\mathrm{CAN}}^k$ is also similar, so we avoid repeating the arguments. The only difference is that we explore all harmonic polynomials of degree at most $k$, instead of $G$-invariants, which corresponds to having larger first term above. But, the second, term corresponding to the approximation error is improved, as we use the alignment hypothesis instead of the Sobolev condition.

*Remark* 13. All these bounds are optimal in minimax sense according to standard bounds in the literature (Bach, 2024; Wainwright, 2019). As the main focus of this paper is not on serving optimal bounds and solely on understanding achievable rates for canonicalization, we avoid rewriting optimality proofs here to keep the paper concise.

$\square$

## E   PROOF OF COROLLARY 7

*Proof.* To derive a formula for the critical sample complexity $N_{\mathrm{critical}}$, we note that we have already shown the following using Lemma 11 to hold with high probability:

$$\left\|\widehat{f}_{\mathrm{CAN}} - f^\star\right\|_{L^2(S^{d-1})}^2 \lesssim \frac{2k^{d-1}}{(d-1)!}\frac{\sigma^2}{n} + k^{-2\beta}, \tag{75}$$

$$\left\|\widehat{f}_{\mathrm{GA}} - f^\star\right\|_{L^2(S^{d-1})}^2 \lesssim \frac{2k^{d-1}}{(d-1)!}\frac{\sigma^2}{|G|n} + k^{-2s}. \tag{76}$$

Therefore, to obtain the phase transition, we solve

$$\frac{2k^{d-1}}{(d-1)!}\frac{\sigma^2}{N_{\mathrm{critical}}} + k^{-2\beta} = \frac{2k^{d-1}}{(d-1)!}\frac{\sigma^2}{|G|N_{\mathrm{critical}}} + k^{-2s}. \tag{77}$$

This leads to

$$N_{\mathrm{critical}} = \frac{\sigma^2 k^{2s+d-1}}{(d-1)!(1-|G|^{-1})(1-k^{2s-2\beta})}. \tag{78}$$

The proof is thus complete.

$\square$

