# OpenReview forum: "Generalization Bounds for Canonicalization: A Comparative Study with Group Averaging"
_ICLR.cc/2025/Conference — ICLR 2025 Poster_

### Official Review · Reviewer_1T9x · 2024-10-29

**Soundness:** 3
**Presentation:** 3
**Contribution:** 3
**Rating:** 6
**Confidence:** 3

**Summary:**

Existing studies have explored the expressive power and continuity of functions represented through canonicalization, but its generalization ability remains underexamined. This paper fills this gap by providing a theoretical analysis of the generalization benefits and sample complexity associated with canonicalization, comparing these attributes with group averaging—a widely used technique for constructing invariant or equivariant function classes.

**Strengths:**

1. This paper fills the gap in theoretical analysis within the field of XX.
2. The paper provides detailed proofs for all theoretical results.
3. The paper demonstrates the validity of the proposed theoretical results through experiments.

**Weaknesses:**

1. The details of the proofs need to be refined, as the reasoning in many steps is not particularly straightforward.
2. As a paper focused on statistical theory, it would be best to outline the structure of the proofs to facilitate understanding.
3. Some definitions seem to be incorrect. For example, on the left of Eq. (2), the function $R(f)$ has no independent variable $x$, but $x$ appears on the right of Eq. (2). The similar problem also exists in several place of this paper.

**Questions:**

See "Weaknesses".

---

> ### Author Response · Authors · 2024-11-28
>
> We thank the reviewer for their positive feedback on our paper, and we apologize for the delay in responding due to personal circumstances.
>
> ---
>
> ### Response to Weakness 1 & 2
> Thank you for raising this issue! We have made small changes to the appendices in the revised version to improve the readability of the proof. Additionally, following the reviewer’s suggestion, we have added a new paragraph at the beginning of the proof of the main result (Appendix D, hihglighted in blue), providing an outline of the proof.
>
> ---
>
> ### Response to Weakness 3
>
> Thank you for pointing this out. We reviewed the paper and made minor changes to improve the notation and correct typos. Regarding the specific point mentioned by the reviewer, we agree that the previous notation was confusing and have updated it as follows:  For any function $f$, we now use $\mathcal{R}[f]$ to denote the result of applying the Reynolds operator to $f$ (i.e., group averaging). Since $\mathcal{R}[f]$  is also a function, it can be evaluated as $\mathcal{R}[f] (x)$, which means the function $\mathcal{R}[f]$ evaluated at $x$. This notation has been updated throughout the paper, including in Equation (2), which was specifically mentioned by the reviewer (highlighted in blue).
>
> ---
>
> We are also pleased to report that, following suggestions from other reviewers, we conducted a new experiment during the rebuttal period. This experiment empirically validated the phase transition proved in the paper regarding the sample complexity of canonicalization. These findings have been added to Section 5.2 of the revised paper (highlighted in blue).
>
> ---
>
> We would greatly appreciate your feedback on the updated version and would like to know if your concerns have been adequately addressed. Thank you!

---

### Official Review · Reviewer_ekUV · 2024-10-30

**Soundness:** 3
**Presentation:** 3
**Contribution:** 3
**Rating:** 8
**Confidence:** 3

**Summary:**

This is a theory-oriented paper focusing on the generalization bounds of canonicalization. Specifically, it introduces the concept of "alignment" to describe the performance differences between canonicalization and group averaging under various conditions. Following the theoretical analysis, a proof-of-concept experiment is presented to support the conclusions.

**Strengths:**

1. The organization and writing of this paper are reader-friendly.
2. The good generalization analysis aids researchers in understanding, designing, and applying canonicalization and group averaging techniques.
3. According to the authors, this is the first theoretical exploration of performance differences between canonicalization and group averaging under varying conditions.

**Weaknesses:**

1. The experiment section would benefit from more case studies.
2. There is no specific description of sample generation in the experiments.

**Questions:**

N/A

---

> ### Author Response · Authors · 2024-11-28
>
> We thank the reviewer for their positive feedback on our paper, and we apologize for the delay in responding due to personal circumstances.
>
> ---
>
> ### Response to Weakness 1
>
> Thanks for raising this issue! To follow the suggestion, we conducted a new experiment to empirically validate our theoretical comparison between canonicalization and group averaging. While our previous experiment focused on the concept of alignment, the new experiment examines the role of sample size in determining the superiority of group averaging versus canonicalization. Specifically, we used point clouds invariant under permutations of their points and compared group averaging with canonicalization (via lexicographic sorting) to derive invariant function classes. Given the computational complexity of averaging over the entire permutation group, we restricted the experiment to point clouds with five points.
>
> Our findings demonstrate that group averaging is superior when the sample size is small, whereas canonicalization outperforms it as the sample size increases. These results align with the theoretical predictions in our paper.
>
> The new findings have been added to the revised version of the paper (Section 5.2, highlighted in blue).
>
>  ---
>
> ### Response to Weakness 2
>
> Thank you for pointing this out! We have revised the paper (Section 5.1) to include additional details about the sample generation process (highlighted in blue).  Specifically, we used $n$ i.i.d. samples from $[-1, 1]^d$ for both the training and test datasets, with $n = 100$ and $d = 3$.
>
>
> ---
>
> We would greatly appreciate your feedback on the updated version and would like to know if your concerns have been adequately addressed. Thank you!

---

> > ### Comment · Reviewer_ekUV · 2024-11-29
> >
> > Thanks for your positive and valuable response, which addressed my concerns. In addition, the concerns of other reviewers, especially cZFY, are worth noting, and I hope you can address them properly.

---

> > > ### Author Response · Authors · 2024-12-02
> > > **Thank you**
> > >
> > > Thank you very much for your positive feedback and constructive comments!

---

### Official Review · Reviewer_cZFY · 2024-10-31

**Soundness:** 3
**Presentation:** 3
**Contribution:** 2
**Rating:** 5
**Confidence:** 3

**Summary:**

The paper theoretically investigates the generalization error of canonicalization vs group averaging. The theoretical results suggest that the generalization performance of canonicalization and group average depends on the trade-off between variance and bias (approximation error), where group averaging being superior in variance (by a factor of group size) and canonicalization being superior in bias based on a newly defined alignment metric. The authors also provide a set of proof-of-concept simulation to show that group averaging and canonicalization outperforms each other under different true functions.

**Strengths:**

1. Theoretical results have consistent notations and are easy to follow.

2. The theoretical results shows the interesting trade-off between bias and variance in the generalization error of canonicalization and group averaging, where canonicalization, in general, has better bias and worse variance. This alludes to the idea that canonicalization might benefit in the case of large data size.

3. The trade-off also results in a critical data size value, which unveils an indicator to compare group averaging and canonicalization.

**Weaknesses:**

1. Corollary 7 seems to be useful and interesting at the first glance, but there are some issues with it. First of all, equation (17) and (18) could be stated more carefully even though the idea is intuitive, as the main theorem provides upper bound rather than precise value (which could make the argument false with small probability). Second, the critical value depends on some factors that are practically intractable, mainly the $\beta$ alignment factor, but also the noise level. Third, in the case of $\beta = s$ as suggested in Proposition 3, the critical value goes to infinity, which suggests canonicalization will always be inferior.

2. The statement of Proposition 3 might also cause confusion. First, it is better to state the function class here to remind the reference function class when talking about alignment. Second, upon further reading into the Appendix C, it seems the statement should be $\beta \geq s$ rather than $\beta = s$ (The authors discussed this relation above and below the paragraph, but it could be better to have it in the statement unless there are concerns the authors like to clarify), as this $s$ derivation is based on group averaging and Theorem 1. Also, when $s=\beta$, it actually causes problem in Corollary 7, where the critical value will go to infinity, which suggests canonicalization will always be worse for G-invariant Sobolov function with parameter s.

3. Simulation is lacking. The provided simulation does align with the general intuition that when the true function is close to the canonicalization representation, canonicalization could do better. However, this simulation does not provide any insights on the effect of data size, which is the main theoretical contribution of the work (the error comparison over the number of observations). I hope the author could at least provide a simple set of simulations that shows the error decay of group averaging vs canonicalization which shows the two converge as n increase and then canonicalization takes the lead. I also suggest investigating the empirical critical value (when error of group averaging starts to surpass canonicalization) over different factors to see whether Corollary 7 could have practical implications for future iterations (considering the limited time for the rebuttal).

1. There are some minor typos that might confuse some readers. For example, one super-script G is missing in equation (15).

**Questions:**

1. The authors mentioned in the literature review that people have studied the generalization error of group averaging for kernel methods. However, how does existing bound compares with the bound derived by the authors is not discussed. Can the author share some details on this matter and remark the novelty here?

2. I have a minor question regarding the example 4 and please correct me if I am wrong. The example is given to demonstrate when the closed space assumption is violated, there could be cases where $F_{GA}$ is not a subset of $F_{CAN}$. However, as stated in line 300, "the function belongs to $F_{CAN}$ but not $F_{GA}$, which suggests $F_{GA}$ is not a subset of $F_{CAN}$." I am not quite following the logic here as I believe we need something on the contrary right?

---

> ### Author Response · Authors · 2024-11-28
>
> We thank the reviewer for their positive feedback on our paper and their recommendation to include additional experiments. We apologize for the delay in responding due to personal circumstances.
>
> ---
>
> ### Response to Weakness 1
>
> Thanks for your comment! First, we agree with the reviewer that Equation 17 and Equation 18 can be stated more formally. Therefore, in the revised version, we followed the reviewer's suggestion and made Corollary 7 more formal (highlighted in blue).
>
> Second, we aggree that the critical value depends on the problem's parameters which can be difficult to estimate in practice. In the paper, our foucs was to show that the phase transition exists, and to show that the critical value depends on the problem's parameters via the given formula, assuming that we have some knowledge of the taks and the model.
>
> Third, the reviewer’s point about there being no gain in canonicalization when $\beta = s$ is exactly correct, as it corresponds to the case where no transition occurs. Intuitively, this situation arises when the canonicalization map introduces unnecessary complications (e.g., non-differentiability) into the problem. As a more concrete example, if the optimal target function is a permutation-invariant linear function on $\mathbb{R}^d$, then performing group averaging over the space of linear functions is indeed superior to canonicalization (e.g., via sorting the vector), since the canonicalized function class is larger (Theorem 1), thereby introducing more variance, while the bias of both group averaging and canonicalization is zero in this case. We have also added a clarifying sentence about this phenomenon in Corollary 7.
>
> ---
>
>
> ### Response to Weakness 2
>
> Thank you for pointing out the ambiguities in the statement of Proposition 3. We would like to clarify that the proposition holds only when $\beta = s$, and not for larger values of $\beta$. To understand this, we note that the definition of Sobolev space involves a specific decay of the spectrum (Equation 31 in the paper), which allows us to conclude the result only for $\beta = s$.
>
> Furthermore, we agree with the reviewer that if $\beta = s$, there is no gain in canonicalization, as in this case no alignment occurs in the problem to any extent.
>
> The statement of Proposition 3 has been revised in the paper to address the reviewer’s concerns (highlighted in blue).
>
> ---
>
>
> ### Response to Weakness 3
>
>
> Thanks for your recommendation to include additional experiments! To follow the suggestion, we conducted a new experiment to empirically validate our theoretical comparison between canonicalization and group averaging. While our previous experiment focused on the concept of alignment, the new experiment examines the role of sample size in determining the superiority of group averaging versus canonicalization. Specifically, we used point clouds invariant under permutations of their points and compared group averaging with canonicalization (via lexicographic sorting) to derive invariant function classes. Given the computational complexity of averaging over the entire permutation group, we restricted the experiment to point clouds with five points.
>
> Our findings demonstrate that group averaging is superior when the sample size is small, whereas canonicalization outperforms it as the sample size increases. These results align with the theoretical predictions in our paper.
>
> The new findings have been added to the revised version of the paper (Section 5.2, highlighted in blue). We also plan to study the dependence of the critical sample complexity on the problem's parameters through experiments, and we will include these results in the next version of the paper (due to time constraints during the rebuttal period).
>
>
>
>
> ---
>
> ### Response to Weakness 4
>
> Thank you for pointing out the typo! We made a pass through the paper and corrected all typos we found, including the one you mentioned, in the revised version.

---

> ### Author Response · Authors · 2024-11-28
>
> ### Response to Question 1
>
>
> Thank you for asking this question. We note that the study of sample complexity for group averaging in kernel methods heavily relies on the concept of sparsity. Kernels, under mild conditions, adopt spectral representations (via, e.g., Mercer's theorem) and often lead to an infinite-dimensional spectral feature vector that becomes sparse when we apply group averaging over the kernel to restrict to invariant functions. Therefore, the key distinction when comparing group averaging with non-invariant kernel methods is whether sparsity is present.
>
> However, in our paper, we focus on comparing canonicalization with group averaging, where the concept of sparsity is no longer sufficient. While we used sparsity to derive the bound for group averaging, for canonicalization, we introduced the notion of alignment, which is fundamentally independent of the sparsity-based approach used in previous works.
>
>
>
> [1] Bietti, Alberto, Luca Venturi, and Joan Bruna. "On the sample complexity of learning under geometric stability." Advances in neural information processing systems 34 (2021): 18673-18684.]
>
> [2] Tahmasebi, Behrooz, and Stefanie Jegelka. "The exact sample complexity gain from invariances for kernel regression." Advances in Neural Information Processing Systems 36 (2023).
>
>
> [3] Elesedy, Bryn. "Provably strict generalisation benefit for invariance in kernel methods." Advances in Neural Information Processing Systems 34 (2021): 17273-17283.
>
> [4] Mei, Song, Theodor Misiakiewicz, and Andrea Montanari. "Learning with invariances in random features and kernel models." Conference on Learning Theory. PMLR, 2021.
>
>
> ---
>
> ### Response to Question 2
>
> Thanks for pointing out this issue! The purpose of Example 4 is to complete what we proved in Theorem 1 about  $F_{CAN} \supseteq F_{GA} $.
>
> In that example, we show that there are cases, that satify the condition in Theorem 1, and we have      $F_{CAN} \supset F_{GA}$ and  $F_{CAN} \neq F_{GA}$. In other words, the example show that there are cases that  $F_{GA}$ is a proper subset of  $F_{CAN}$. We made changed to clarify this in the revised version (paragraph before Example 4, highlighted in blue).
>
> ---
>
> We would greatly appreciate your feedback on the updated version and would like to know if your concerns have been adequately addressed. Thank you!

---

> > ### Author Response · Authors · 2024-12-02
> > **Looking Forward to Your Feedback**
> >
> > Dear Reviewer cZFY, as the discussion period is about to conclude, we are wondering if there are any additional concerns we should address that could lead to a more positive evaluation of our work in your view.
> >
> > Upon your request, in our revised version and over the past few weeks, we performed a **new experiment** demonstrating the phase transition in test error for group averaging vs. canonicalization, aligning with our proposed theory. This has been added to the revised version (Section 5.2, highlighted in blue).  We have also provided a point-by-point response to your other concerns in our previous comments.
> >
> > We are happy to clarify or improve any aspects of our work and engage in further discussions before you finalize your rating. Thank you!

---

### Official Review · Reviewer_JVqd · 2024-11-05

**Soundness:** 3
**Presentation:** 3
**Contribution:** 3
**Rating:** 5
**Confidence:** 2

**Summary:**

This paper addresses this gap by theoretically examining the generalization benefits and sample complexity of canonicalization, comparing them with group averaging, another popular technique for creating invariant or equivariant function classes.

**Strengths:**

It provides new insights into the generalization bounds of canonicalization and enriches related theoretical research. By comparing with the group averaging method, it highlights the advantages and disadvantages of different methods and provides practical references for researchers. The paper has a reasonable structure, clear logic, and is easy for readers to understand. The paper studies topics of great significance in machine learning and statistics, which is in line with current research trends.

**Weaknesses:**

Although the theoretical analysis is thorough enough, it might be helpful to add some experiments to ensure empirical support.

**Questions:**

none

---

> ### Author Response · Authors · 2024-11-28
>
> We thank the reviewer for their positive feedback on our paper and their recommendation to include additional experiments. We apologize for the delay in responding due to personal circumstances.
>
> To follow the suggestion, we conducted a new experiment to empirically validate our theoretical comparison between canonicalization and group averaging. While our previous experiment focused on the concept of alignment, the new experiment examines the role of sample size in determining the superiority of group averaging versus canonicalization. Specifically, we used point clouds invariant under permutations of their points and compared group averaging with canonicalization (via lexicographic sorting) to derive invariant function classes. Given the computational complexity of averaging over the entire permutation group, we restricted the experiment to point clouds with five points.
>
> Our findings demonstrate that group averaging is superior when the sample size is small, whereas canonicalization outperforms it as the sample size increases. These results align with the theoretical predictions in our paper.
>
> The new findings have been added to the revised version of the paper (Section 5.2, highlighted in blue).
>
> We would be happy to hear your feedback on the updated version and to know if your concerns about the experiments have been adequately addressed. Thanks!

---

> > ### Author Response · Authors · 2024-12-02
> > **Looking Forward to Your Feedback**
> >
> > Dear Reviewer JVqd, as the discussion period is about to conclude, we are wondering if there are any additional concerns we should address that could lead to a more positive evaluation of our work in your view.
> >
> > Upon your request, in our revised version and over the past few weeks, we performed a **new experiment** demonstrating the phase transition in test error for group averaging vs. canonicalization, aligning with our proposed theory. This has been added to the revised version (Section 5.2, highlighted in blue).
> >
> > We are happy to clarify or improve any aspects of our work and engage in further discussions before you finalize your rating. Thank you!

---

### Meta-Review · Area_Chair_c7Gt · 2024-12-20

**Metareview:**

Thanks for your submission to ICLR.

This paper received borderline reviews, with two positive and two negative reviews.  The major issues raised by the reviewers include: i) additional requested experiments, ii) various issues with the theoretical results.

The authors responded to these issues and, during the discussion phase, there was somewhat of a consensus amongst the reviewers that the paper was strong enough for publication.  In particular, reviewer cZFY, who had many of the concerns about the theoretical results, indicated that they felt the paper could be published.  They did not actually raise their score (so the score is somewhat misleading), but I will paste some text that this reviewer made during discussion:

"I think the authors have made a satisfactory response and revision to my questions, and the added experiment in section 5.2 also verifies the existence of the transition phase, which is the main theoretical contribution of this paper. Although how to identify the critical point of transition is alluded by the authors but not really demonstrated, the proof of existence of such transition point is, in my opinion, already a good contribution. Therefore, I would recommend acceptance."

The other negative reviewer did not respond during discussion, but their primary concern (of having some experiments added) was adequately addressed during the rebuttal and discussion phase.

Thus, I am happy to go ahead and recommend accepting this paper.

**Additional Comments On Reviewer Discussion:**

See above.  One of the two negative reviewers ultimately recommended acceptance (but did not change their score), and the other negative reviewer's concerns were addressed adequately.

---

### Decision · Program_Chairs · 2025-01-22

Accept (Poster)